# Behavioral Testing of Knowledge Graph Embedding Models

**Wiem Ben Rim**[1]                                    WIEM.BENRIM@NLP.C.TITECH.AC.JP
**Carolin Lawrence**[2]                                CAROLIN.LAWRENCE@NECLAB.EU
**Kiril Gashteovski**[2]                               KIRIL.GASHTEOVSKI@NECLAB.EU
**Mathias Niepert**[2]                                 MATHIAS.NIEPERT@NECLAB.EU
**Naoaki Okazaki**[1]                                  NAOAKI.OKAZAKI@C.TITECH.AC.JP
[1] *Okazaki Laboratory, Tokyo Institute of Technology, Tokyo, Japan*
[2] *NEC Laboratories Europe, Heidelberg, Germany*

## Abstract

Knowledge graph embedding (KGE) models are often used to encode knowledge graphs in order to predict new links inside the graph. The accuracy of these methods is typically evaluated by computing an averaged accuracy metric on a held-out test set. This approach, however, does not allow the identification of *where* the models might systematically fail or succeed. To address this challenge, we propose a new evaluation framework that builds on the idea of (black-box) behavioral testing, a software engineering principle that enables users to detect system failures before deployment. With behavioral tests, we can specifically target and evaluate the behavior of KGE models on specific capabilities deemed important in the context of a particular use case. To this end, we leverage existing knowledge graph schemas to design behavioral tests for the link prediction task. With an extensive set of experiments, we perform and analyze these tests for several KGE models. Crucially, we for example find that a model ranked second to last on the original test set actually performs best when tested for a specific capability. Such insights allow users to better choose which KGE model might be most suitable for a particular task. The framework is extendable to additional behavioral tests and we hope to inspire fellow researchers to join us in collaboratively growing this framework. The framework is available at https://github.com/nec-research/KGEval.

## 1. Introduction

Knowledge Graphs (KGs) are graph databases that represent information about entities and their relationships in the form of canonical (head, relation, tail)-triples. KGs are used in many downstream tasks such as question answering [Huang et al., 2019], recommender systems [Guo et al., 2020], information extraction [Gashteovski et al., 2020], and named entity linking [Shen et al., 2012]. A common challenge when working with KGs is that they suffer from incompleteness [West et al., 2014]. For example, the popular KG DBPEDIA [Lehmann et al., 2015] contains information about the entities Joe Biden and United States, but it does not contain the fact expressing the presidentOf relationship between the two entities. Motivated by the incompleteness of most KGs, there is a large body of work on link prediction in knowledge graphs, that is, on deriving missing triples from the set of existing triples.

KG Embeddings (KGE) have been shown to be effective at predicting missing links [Minervini et al., 2015, Ruffinelli et al., 2020]. KGE methods learn low-dimensional representations of entities and relation types in a vector space (see also the more detailed definition

in Section 2.1). KGE models are typically evaluated by measuring their ability to rank candidate entities averaged over triples in a held out test set. This average-based evaluation, however, poses several problems (Bianchi et al. [2020], Kadlec et al. [2017], Sun et al. [2020b], Mohamed et al. [2020]; for a closer discussion see also Section 2.2) and leaves open the question of *what* models fail to learn.

As a step towards a better evaluation of KGE models, we propose the use of behavioral tests. Behavioral testing, a standard-practice in software engineering, is concerned with testing behaviors of a (black-box) software system by feeding it various inputs and observing and analyzing the system's behavior. Such behavioral tests have several advantages. First, it is possible to find out if a model makes systematic mistakes for a certain capability of interest. Second, one can compare different models in a more detailed and fine-grained manner, allowing us to understand under which circumstances different models offer an advantage. Third, the tests could be used to uncover particular issues in the training data, which could then be corrected (e.g. in the context of KGs by adding more entities or relations of a certain type). Fourth, by using a more rigorous testing approach, we can increase the trust of stakeholders in production settings.

In this work, we kick-start a new evaluation framework by exploring possible testable capabilities and by defining detailed tests for two of these capabilities. First, we explore how well KGE models handle symmetric relations (e.g. the relation `spouseOf` is symmetric), which has been a much discussed property (Sun et al. [2019], Peng and Zhang [2020], Zhang et al. [2020], Trouillon et al. [2016], Wang et al. [2014]; *inter alia*) but so far lacks a systematic evaluation. Second, entities in a KG are often associated with an entity type (e.g. `Diane Sawyer` is of type ACTOR) and we evaluate how well KGE models have learnt to respect entity types. We run these tests on six KGE models and find that the model ranking on the original test set does not reflect the same ranking when testing for specific capabilities. For instance, we find that the model COMPLEX [Trouillon et al., 2016] ranks second to last on the original test set but is the best at predicting unseen triples for symmetric relations.

## 2. Link Prediction for Knowledge Graphs

A knowledge graph $\mathcal{K}$ consists of a set of entities $\mathcal{E}$, a set of relation types $\mathcal{R}$, and a set of triples $d = (h, r, t)$, with head $h \in \mathcal{E}$, relation type $r \in \mathcal{R}$ and tail $t \in \mathcal{E}$. For example, the triple (`Porto`, `locatedIn`, `Portugal`) represents the information that the entity `Porto` is located in the entity `Portugal`. Knowledge graphs are typically incomplete. Link prediction is the task of inferring new triples based on the triples contained in the knowledge graph. This problem can be framed as a tail and head prediction query of the form $(h, r, ?)$ and $(?, r, t)$, where one seeks to find substitutions for ? that result in a new correct triple.

### 2.1 Knowledge Graph Embedding Models

A Knowledge Graph Embedding (KGE) model consists of three main components. First, the KGE model's parameters $\boldsymbol{w}$. Typically, the parameters are vectors associated with each entity and relation type. Second, given the model's parameters $\boldsymbol{w}$ we have a scoring function $\phi(d; \boldsymbol{w})$ which maps the parameters of the head, tail, and relation type occurring in a triple $d$ to a real-valued number. KGE models differ mainly in the particular choice of

scoring functions. Third, the model's parameters are learnt based on a training set $\mathcal{D}_{train}$ of known triples, where a commonly used loss function aims to maximize the score of known triples while minimizing the score of randomly sampled (much more likely to be incorrect) triples. Once a model is trained, it can predict a score for an unseen triple that indicates the likelihood of this being true.

## 2.2 Standard Evaluation Metrics and their Shortcomings

The standard approach for evaluating link prediction methods uses a (held-out) test set $\mathcal{D}_{test}$ of correct triples. For each of these $(h, r, t)$-triples, the model scores all possible substitutions of the queries $(h, r, ?)$ and $(?, r, t)$. Then, the entities are ranked in descending order according to their predicted scores. There are two commonly used metrics for evaluating the resulting rankings—MRR and Hits@k ($t$ may be replaced with $h$ for the reverse direction):

$$\text{MRR} := \frac{1}{|\mathcal{D}_{test}|} \sum_{(h,r,t) \in \mathcal{D}_{test}} \frac{1}{\texttt{rank}(t)}; \qquad \text{Hits@k} := \frac{1}{|\mathcal{D}_{test}|} \sum_{(h,r,t) \in \mathcal{D}_{test}} \mathbb{1}[\texttt{rank}(t) \leq k].$$

The MRR (mean reciprocal rank; lower is better) metric identifies the gold tail $t$'s rank $\texttt{rank}(t)$ according to the model's scores and computes the mean of the reciprocal ranks over all test triples. The Hits@k metric computes the mean, over all test triples, of the event that the gold tail occurs in the top $k$ ranked entities.

Using the above metrics as averages over a test set is a global and coarse-grained measure of accuracy. It conflates different relation and entity types and does not allow the user to understand the types of errors the KGE model makes on particular groups of triples. For example, the model could perform poorly on specific relation types, either due to a weakness of the model or misspecified training data. The need for an evaluation approach that can analyze and explore the behavior of KGE methods in a more fine-grained manner was also expressed in prior work [Bianchi et al., 2020]. To move towards a better evaluation paradigm for KGE models, we propose a new, extendable evaluation framework which can compare KGE models in a more systematic manner.

In addition to the coarse-grained nature of standard evaluation measures, prior work has identified additional shortcomings of common evaluation procedures. First, Kadlec et al. [2017] found that hyperparameter tuning can be more important than model architecture changes, which caused the authors to raise doubts about the traditional accuracy based evaluation. By using and extending our framework of more fine-grained behavioral testing, it will be possible to determine if and under which circumstances different model architecture perform better. Second, it has been found that recently published (and seemingly superior) KGE models have been evaluated incorrectly [Sun et al., 2020b]. By introducing a common test framework, such inconsistency could be avoided. Third, Mohamed et al. [2020] has shown that the traditional accuracy-based metrics overestimate the performance of KGE models because they magnify the accuracy on frequently occurring entities and/or relations. The less frequent, "long-tail" triples, however, are often the triples that one cares the most about [Ilievski et al., 2020].

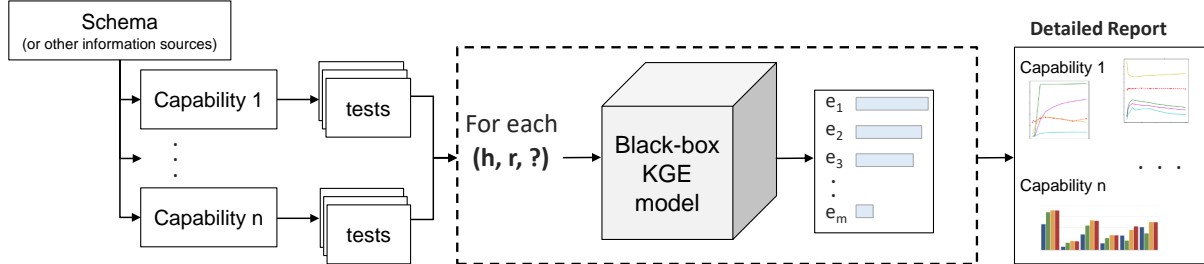

Figure 1: An overview of the proposed framework to analyze KG embedding (KGE) models and to explore systematic failure modes. Given a KG's schema (or other external sources), we can define various system capabilities. To test a capability, various test sets with relevant triples are created. For each triple in each test set, we can query the KGE model. The KGE model can be a black-box, which, given a link prediction query $(h, r, ?)$, only outputs a score for each possible entity. Based on this, a detailed report about the model's test behavior is provided.

## 3. Behavioral Testing

**Overview.** We propose to use behavioral tests as a new evaluation framework that allows one to better understand *what* KGE models succeed or fail to learn. Behavioral testing is a software engineering principle where capabilities of software systems are tested by treating them as black boxes and analyzing their behavior for specifically designed inputs. For example, a software engineer might write a function that expects positive integers inputs. One behavioral test for this function could be a test that checks what happens if the function is given a negative integer instead - does the function handle this disallowed case appropriately?

The idea of behavioral testing has also recently been applied to NLP [Ribeiro et al., 2020]. For example, Ribeiro et al. [2020] analyzed the robustness of named entity recognition systems by replacing the city name in specific sentences with different city names, and observing the resulting change in behavior of the system. Here, we transfer the idea of behavioral testing to analyze the behavior of KGE models in a more fine-grained manner.

Knowledge graphs are typically associated with a predefined schema [Auer et al., 2007], which imposes constraints on the set of possible triples. For instance, a given relation type might have entity type constraints for its domain and range (e.g., the DBpedia relation type `birthPlace` must have an entity of type `person` as head and an entity of type `location` as tail). We propose to use such schemas to design behavioral tests and to check the consistency of KGE models with regards to the schema constraints.

Based on a given KG schema, fine-grained tests can be designed to assess KGE models with respect to their behavior for particular capabilities. As a result, this type of evaluation can offer more detailed insights into where KGE models perform well and where they might make systematic mistakes. A graphical overview of our proposed approach is depicted in Figure 1.

**Possible Capabilities.** Our framework can be easily extended by adding new capabilities and corresponding tests. Adding various such capabilities and tests to one joint evaluation

framework would ensure better comparison between different KGE models in the future. Some possible capabilities are:

- **Relation symmetry:** KGs often contain symmetric relations, where a relation $r \in \mathcal{R}$ is symmetric if $\forall x, y \in \mathcal{E}, (x, r, y) \implies (y, r, x)$. For example, if a KGE model was trained on the symmetric relation (x, spouseOf, y), a test could check if it correctly predicts (y, spouseOf, x). Ensuring that a KGE model can handle symmetric relations has been the focus of several recently proposed KGE models (Sun et al. [2019], Peng and Zhang [2020], Zhang et al. [2020], Trouillon et al. [2016], Wang et al. [2014]; *inter alia*). The idea can also be extended to other properties of relation types, such as antisymmetry, inversion, and composition.

- **Entity hierarchy:** Entities can be associated with an entity type. For example, the FB15K-237 entity Diane Sawyer is of type ACTOR. The entity types themselves are organized in a hierarchical (hypernym) taxonomy; e.g. ACTOR → ARTIST → PERSON. Various KGE models have specifically been designed to learn entity hierarchies well [Zhang et al., 2020, Kolyvakis et al., 2020, Balazevic et al., 2019, Chami et al., 2020, Sun et al., 2020a]. Tests could for example explore how well KGE models work at different hierarchy levels.

- **Entity distributions:** Relations can be grouped into 4 categories with regards to how many correct heads/tails they may have: 1-TO-1, 1-TO-MANY, MANY-TO-1 and MANY-TO-MANY. For example, some relations by definition can only have one correct tail as an answer (e.g. birthPlace). Prior work studied how well KGE models perform in such different scenarios (e.g.[Bordes et al., 2013a, Peng and Zhang, 2020]) and this type of analysis could easily be added to the evaluation framework.

- **Robustness to adversarial attacks:** Pezeshkpour et al. [2019] study how KGE models are effected by adversarial modifications. This idea could be transformed into a capability that checks how prone KGE models are to adversarial attacks.

- **Relation/entity frequency:** The observation that KGE models perform better on frequently occurring entities/relations [Mohamed et al., 2020] could be systematically tested by creating different test sets where this frequency is varied.

The tests defined for a capability explore the behavior of a model under a particular setting or condition. They can be utilised in two ways. First, they can serve as evaluation benchmarks with which different systems can be compared to each other. Based on this, we can choose the best model for a particular capability of interest. Second, they can be used to determine the failure rate of a system by making a binary decision for each triple, e.g. by defining a cut-off point for each tested triple.[1] With this view, we can determine if a model is good enough to be deployed. While the latter is ultimately more important in a production setting, we focus here on the former, with which we explore how known KGE models can be compared against each other.

---

1. For example, for a system to succeed a triple's gold tail has to be in the top 3 of a model's prediction, else it is counted as failure.

**Evaluation Setup.** To kick-start the new evaluation framework, we define several tests for two capabilities and then use the tests to evaluate different KGE models. For the evaluation, we employ six KGE models: DistMult [Yang et al., 2015], ComplEx [Trouillon et al., 2016], RotatE [Sun et al., 2019], HyperKG [Kolyvakis et al., 2020], LinearRE [Peng and Zhang, 2020], HAKE [Zhang et al., 2020]. For an overview of their scoring functions, see Table 1 in the appendix.

The first capability we explore is relation symmetry. All proposed methods can, in principle, learn that a relation type is symmetric. DistMult in particular treats all relation types as such. However, the main evaluation for these models was conducted on the standard test sets with averaged metrics. While ablation studies regarding symmetric capability are sometimes included [Sun et al., 2019, Peng and Zhang, 2020],[2] they are neither systematic nor comparable across papers. Therefore, it is difficult to judge to what degree they can handle symmetry and which model might be the best with regards to this capability. To be able to answer this question, we design targeted tests which allow us to measure the performance specifically for symmetric capability. In turn, we can analyse in detail under which settings which model performs best.

The second capability we investigate is entity hierarchy. Many KGs are hierarchical by nature; as an example see Figure 3 for entity types in DBpedia. This encodes important additional knowledge; for instance, certain relations only apply to certain entity types. For example, the relation starsIn is typically associated with tail entities of type Actor, which is at level 4 in the hierarchy (see Figure 3). In contrast, the relation birthPlace can have tail entities of the higher ranked level 1 type Place (e.g. City or Town, because both entity types are a type of Place). For this capability, we design tests that explore how well KGE models have learnt to handle different entity types at varying levels in the entity type hierarchy. For this capability, we can test whether KGE models that have specifically been designed to learn hierarchical structure (HyperKG, HAKE) perform better than other models.

The models are trained on FB15k-273 [Toutanova et al., 2015] and we evaluate the models which achieved highest performance on the original FB15k-273 test set. All hyperparameters are included in Section C of the appendix.

### 3.1 Capability: Relation Symmetry

Some relations in a knowledge graph are symmetric: given $x, y \in \mathcal{E}$ and $r \in \mathcal{R}$, if $(x, r, y)$ is true, then $(y, r, x)$ is also true and vice versa. We define four tests to better understand the extent to which KGE models have the ability to handle symmetry. The first three tests target symmetric relations and become progressively more difficult. The last test examines how well models recognize if a relation is not symmetric (or asymmetric).

To set up these tests, the relations of a dataset need to be split into symmetric and asymmetric relations. For FB15k-237 we report the set of symmetric relations in Table 2 in the appendix. All other relations of the dataset are asymmetric. Based on the set of symmetric relations, we find 6,220 symmetric triples in the training set and 2,520 triples

---

2. Sun et al. [2019] supply some example relations where RotatE identifies a relation to be symmetric/asymmetric; Peng and Zhang [2020] note that RotatE performs better than TransE [Bordes et al., 2013b] on some datasets because they contain more symmetric relations.

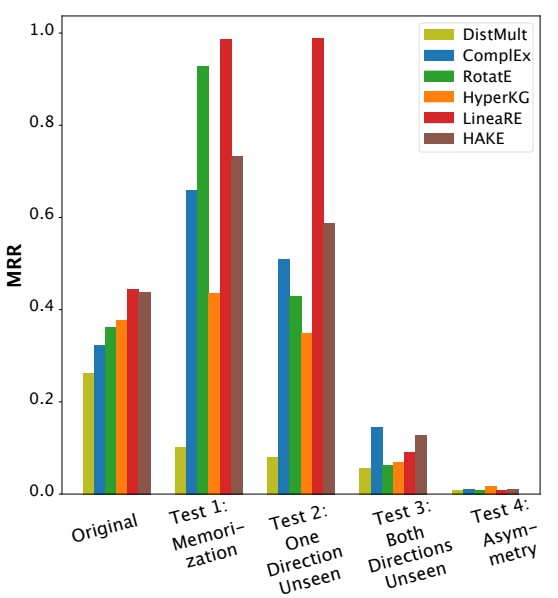

(a) MRR results of all models on the original and relation symmetry test sets.

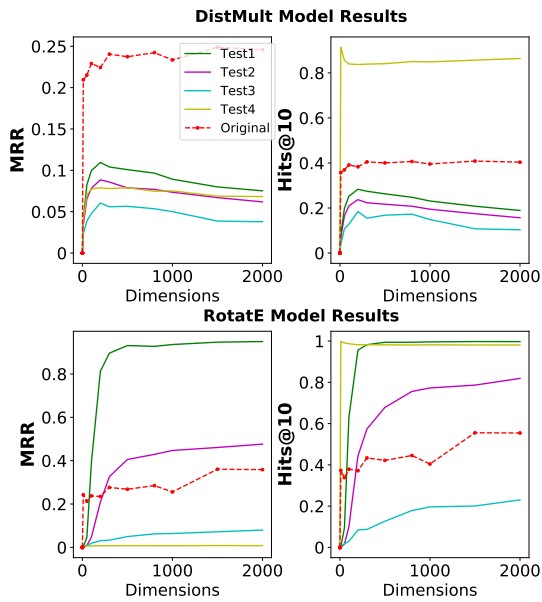

(b) MRR and Hits@10 results for DISTMULT and ROTATE for different dimension sizes.

Figure 2: Results for relation symmetry tests. (a) shows a clear disparity between the different tests. Most interestingly, COMPLEX is ranked second to last on the original test set but performs best when testing unseen triples (Test 3). (b) Both models perform well on Test 4, with ROTATE less likely to mistake an asymmetric relation for a symmetric one.

that occur in both directions. With this knowledge we can now define and instantiate the four tests.

**Test 1: Memorization.** The first and easiest test measures the extent to which a model memorizes training set triples with a symmetric relation type. For each training triple $(x, r, y)$, if $r$ is in the set of symmetric relations, we add $(x, r, y)$ to the test set for Test 1 and ask the model to predict the tail entity. This test can be considered an upper bound for the other tests since the data has been seen during training; thus this test constitutes the easiest scenario.

**Test 2: One Direction Unseen.** For the second test, we create a test set where a triple with a symmetric relation was seen in one direction during training but the reverse direction was not seen: if $(x, r, y)$ is in the training set, but $(y, r, x)$ is not, then $(y, r, x)$ is added to the test set for Test 2. As these triples are unseen, the test is harder than the previous one and it will allow us to investigate how well a model has recognized that a relation is symmetric.

**Test 3: Both Directions Unseen.** The third test consists of triples that have a symmetric relation but were never seen in either direction during training. For this test we collect unseen triples from the validation and test set and if a relation $r$ is symmetric, we

add both $(x, r, y)$ and $(y, r, x)$ to the test set for Test 3. Since neither direction was seen by the model, this test is more difficult than the previous two and measures how well symmetric relations generalize to unseen triples.

**Test 4: Asymmetry.**  The last test aims to analyze whether a model mistakenly considers a relation symmetric. For instance, for the triple (`George W. Bush, fatherOf, George H. W. Bush`), if the model is given the instance with the head and tail inverted: (`George H. W. Bush, fatherOf, ?  `), then `George W. Bush` should not be among the top predictions. This implies that a higher MRR is worse for this test. To probe the behavior of KGE models with respect such asymmetry, we randomly sample triples from the training set that do not contain a symmetric relation.

**Results.**  Results on the original test set of FB15k-237 and our behavioral tests are shown in Figure 2 (a). On the original test set LineaRE and HAKE perform best and obtain comparable MRR. However, the behavioral tests expose interesting differences between the two models. LineaRE performs far better than HAKE on Test 1 ("*Memorization*") and Test 2 ("*One Direction Unseen*"). This indicates that LineaRE is better at memorizing the training set (Test 1) and recognizing symmetric relations (Test 2). However, on the the test that measures generalization to unseen triples (Test 3, "*Both Direction Unseen*"), we find that HAKE outperforms LineaRE. On Test 4 ("*Asymmetry*") higher MRR indicates a worse model. We find that LineaRE and HAKE, along with the other models, perform well on this test, indicating that they are not likely to mistake an asymmetric relation for a symmetric one.

Even more surprising are the results of ComplEx. On the original test set it is ranked second to last (i.e., only DistMult is worse), however, on Test 3 it achieves the best performance. It also performs better than expected (based on the original test set) for Tests 1 and 2. This indicates that if generalizing well on symmetric relations is important, ComplEx would be a promising model to try out. Based on these results, additional future investigation into why ComplEx performs better than expected on Test 3 could also lead to new insights on how to achieve better symmetric capability.

There are two further interesting results. First, DistMult, which treats all relations as symmetric, performs worst. Second, RotatE performs very well on Test 1 but not on the other tests, which indicates that RotatE can memorize well, but does not generalize to more difficult setups. We investigate this further in Figure 2 (b), where we plot the results of DistMult and RotatE and we evaluate the models for different dimension sizes. For the first three behavioral tests DistMult is better at lower dimension sizes, whereas for RotatE higher dimensions sizes bring a small performance increase. For Test 4, we find that all models in Figure 2(a) perform well (the lower the MRR, the better), showing that models do not easily mistake asymmetric relations for symmetric ones.

## 3.2 Capability: Entity Hierarchy

Entities in a KG may be associated with entity types, which can be expressed in a hierarchical structure. As shown in Figure 3, entity types vary from general and ambiguous concepts, such as Thing or Agent, to more specific ones such as Actor and Ballet Dancer. Therefore, entities are associated with types belonging to different hierarchy lev-

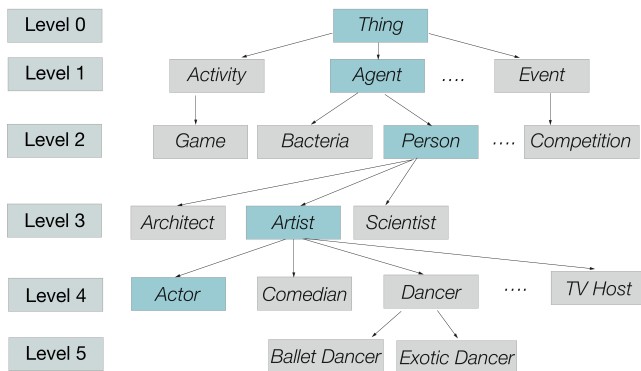

Figure 3: Hierarchy of entity types in DBpedia. Each entity has a entity type.

els. Likewise, KG relations sometimes impose constraints on the entity types about their domain and range (e.g., `birthPlace` accepts as head and tail an entity of type PERSON and PLACE respectively).

We define two tests to analyze to what extent KGE models have learnt to respect entity type constraints when making predictions. The first test explores model performance based on the entity level of gold tails to investigate how well the models handle different specificity levels. In the second test we explore how much model performance could be improved if the model had learnt to associate a relation with the correct entity type. To set up the tests, we map each entity of a triple from FB15K-237 to its counterpart in DBPEDIA, which defines entity types and arranges them in a hierarchical manner (see Figure 3). A few entities have no direct mapping and are thus filtered out, we also filter out level 5 entities as they only occur rarely (21 triples in total).

**Test 1: Gold Tail.** To test how well KGE models perform at different entity type levels, we create a test set for each level. For this, we iterate over each triple in the original test set and look up the level of the gold tail's entity type. The triple is then added to the test set of the found level. With this, we can test how performance varies as we move from very general entity types (Level 0) to more specific ones (Level 4).

**Test 2: Type Constraints.** Next, we would like to test how model performance changes if we explicitly restrict the set of possible tail entities based on entity type. If this improves the results, then this indicates that the models have not yet sufficiently learnt to associate a relation with the correct entity type. To test this, we use the training data to compile for every relation the most likely type at each level. For each test set (Level 0 - 4), we apply entity type restrictions in the following way: (1) At prediction time, given a relation, we restrict the entity set to have the entity type that occurred most often for this relation in the training set. For example, for the relation `BirthPlace` at level 2, we restrict the entity set to be of type PLACE or a subtype of PLACE. (2) We move up in the hierarchy tree, which causes the restriction to relax with each step up. Moving up to level 0 would then be equal to not placing any restrictions.

**Test 3: Gold Type** Finally, we would like to test if a model has learnt to recognize that the entity to be predicted should be of a certain type. For instance, for the triple (`George W. Bush, FatherOf, ?`), the predicted tail should be of type PERSON. To investigate this, we mark the first entity that shares the same type as the gold tail as correct, even if it is not necessarily the gold tail itself. In the previous example, if the correct answer is ranked 300, meaning that the model only obtains a tail of type PERSON at rank 300, then the model clearly struggles with entity types. However, if the model always predicts the correct type as the first prediction, then the model has successfully learned which type of entity to expect.

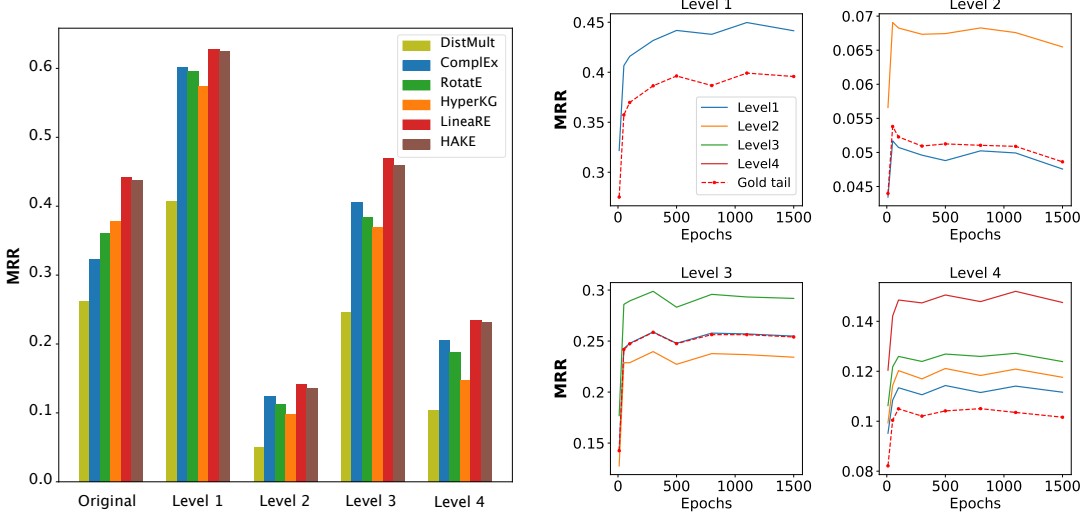

(a) Test 1 results of all models on the original test set and the different entity levels.

(b) MRR results for Test 2, where entity types are constrained with DISTMULT at different epochs.

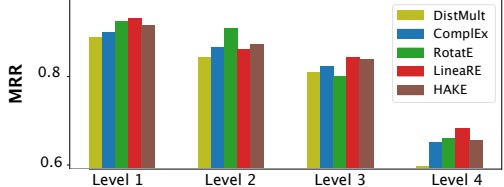

(c) Test 3 results, where a prediction is considered correct if it is of the same entity type as the gold tail.

Figure 4: Results of the hierarchy tests: Figure (a) shows that the difficulty of predicting a tail differs by their level in the hierarchy. Figure (b) shows that models benefit from type constraints. Figure (c) shows that it is easier to predict more general entity types.

**Results.** For the Test 1 ("*Gold Tail*"), Figure 4 (a) shows the performance for the different entity type levels. The model rankings are consistent across the different levels with

LineaRE and HAKE performing best. Regarding the different levels, we find that level 1 entity types are the easiest to predict, while more specific levels seem to be more difficult. In particular, all models perform by far the worst at level 2. The drop of performance in level 2 compared to the other levels likely stems from the type of relation. The relations that occur in level 2 rarely occur in other levels and most of the relations expect a tail of the level 2 type "Person". We conjecture that the models are particularly bad at predicting in this scenario because of the large number of entities that have the type "Person" (4,950; which constitutes 34% of all the entities in the set).

The results for Test 2 ("*Type Constraints*") can be found in Figure 4 (b). We see an improvement in MRR across the board when using type constraints during prediction. For example, for the level 1 test set, we find an improvement of about 0.05 MRR when restricting the entity set to belong to a relation's most common level 1 entity type. Additionally, at all levels, we find that applying the restriction at the same level helps the most. This shows that models have not yet fully learnt to associate entity types with relations. One future direction could be to to investigate how entity type information can be used to further improve models.

Test 3 results are depicted in Figure 4 (c).[3] We find that (1) all models perform similarly across the different levels; (2) at all levels and for all models we observe relatively high MRR ($\geq 0.6$) showing that all models are quite good at correctly predicting the type of the tail entity; (3) performance slightly drops when moving to deeper levels, which is understandable as entities at higher levels are more general and more common; therefore easier to predict (for example, level 1 contains entities of types, e.g. Agent, Deity, while deeper levels are more specific, e.g. Artist or Musician).

These results of Test 3 contrast with Test 2, where we find that despite correctly predicting the correct type at level 2 and level 4, it is much more difficult to predict the correct tail. In the future, we would like to investigate the reason for this discrepancy.

## 4. Conclusion

Knowledge graphs and knowledge graph embedding (KGE) models can be used for link prediction to infer new triples. However, they are typically evaluated using averaged accuracy metrics computed over a test set. As a result, it remains unclear what exactly these models have learnt and which model might be the most suitable for a particular task. We presented a framework to systematically test the capabilities of KGE models using behavioral tests. For two initial capabilities we defined several tests and ran them for six KGE models. Crucially, we find that the model performance on the original test set does not necessarily mirror the same performance when testing models for a specific capability. For instance, CompLEx ranked second to last on the original dataset but is the best at predicting unseen triples for symmetric relations. We hope that this initial framework will also inspire fellow researcher to contribute by adding more tests, models and datasets.

---

3. We were not able to confirm the results for the model HyperKG due to its incompatible implementation for this test.

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

## Appendix A. Scoring Functions

The scoring functions for various KGE models can be found in Table 1.

| Model | Scoring Function $\phi(\boldsymbol{w}; d)$ | Parameters |
|---|---|---|
| DISTMULT | $\langle \boldsymbol{h}, \boldsymbol{r}, \boldsymbol{t} \rangle$ | $\boldsymbol{h}, \boldsymbol{r}, \boldsymbol{t} \in \mathbb{R}^k$ |
| COMPLEX | $\mathrm{Re}(\langle \boldsymbol{h}, \boldsymbol{r}, \bar{\boldsymbol{t}} \rangle)$ | $\boldsymbol{h}, \boldsymbol{r}, \boldsymbol{t} \in \mathbb{C}^k$ |
| ROTATE | $\|\boldsymbol{h} \circ \boldsymbol{r} - \boldsymbol{t}\|^2$ | $\boldsymbol{h}, \boldsymbol{r}, \boldsymbol{t} \in \mathbb{C}^k, |r_i| = 1$ |
| LINEARRE | $\|\boldsymbol{w}_r^1 \circ \boldsymbol{h} + \boldsymbol{b}_r - \boldsymbol{w}_r^2 \circ \boldsymbol{t}\|$ | $\boldsymbol{w}_r^1, \boldsymbol{h}, \boldsymbol{b}_r, \boldsymbol{w}_r^2, \boldsymbol{t} \in \mathbb{R}^k$ |
| HAKE | $-\|\boldsymbol{h}_m \circ \boldsymbol{r}_m - \boldsymbol{t}_m\|_2 -$ | $\boldsymbol{h}_m, \boldsymbol{t}_m \in \mathbb{R}^k, \boldsymbol{r}_m \in \mathbb{R}_+^k$ |
| | $\lambda \|\sin((\boldsymbol{h}_p + \boldsymbol{r}_p - \boldsymbol{t}_p)/2)\|_1$ | $\boldsymbol{h}_p, \boldsymbol{r}_p, \boldsymbol{t}_p \in [0, 2\pi)^k, \lambda \in \mathbb{R}$ |

Table 1: Definitions of the scoring function $\phi(\boldsymbol{w}; d)$ for different models; $\boldsymbol{w}$ the weights of the model, $d = (h, r, t)$ the triple to be scored, $k$ indicates the hidden dimension size, $\langle \rangle$ the inner product, $\mathrm{Re}()$ the real value of a complex vector and $\bar{\boldsymbol{t}}$ the complex conjugate.

## Appendix B. FB15k-237 Symmetric Relations

We report the manually extracted symmetric relations for FB15k-237 in Table 2.

| FB15k-237 Relations |
|---|
| /base/popstra/celebrity/breakup. /base/popstra/breakup/participant |
| /base/popstra/celebrity/canoodled. /base/popstra/canoodled/participant |
| /base/popstra/celebrity/dated. /base/popstra/dated/participant |
| /base/popstra/celebrity/friendship ./base/popstra/friendship/participant |
| /celebrities/celebrity/celebrity_friends. /celebrities/friendship/friend |
| /celebrities/celebrity/sexual_relationships. /celebrities/romantic_relationship/celebrity |
| /influence/influence_node/peers. /influence/peer_relationship/peers |
| /location/location/adjoin_s. /location/adjoining_relationship/adjoins |
| /people/person/spouse_s./people/marriage/spouse |
| /people/person/sibling_s./people/sibling_relationship/sibling |

Table 2: FB15k-237 symmetric relations.

## Appendix C. Hyperparameters of Trained Models

We train models for DISTMULT [Yang et al., 2015], ROTATE [Sun et al., 2019], LINEARRE [Peng and Zhang, 2020] and HAKE [Zhang et al., 2020]. These models differ in their definition of the scoring function $\phi(\boldsymbol{w}; d)$, which may be found in Table 1.

We use the python framework Pykeen [Ali et al., 2021] for all our experiments, it provides ready-to-use implementations of both the DISTMULT and the ROTATE models.

For the DISTMULT experiments, we fix the following hyperparameters: 500 negative per positive samples for the negative sampler, a learning rate of $1e^{-3}$. We train the first group of models for 100 epochs while varying the size of embedding dimensions. For the second group, we train the models for different epochs while keeping the hidden dimensions at 100.

| | | Triples in training set* | | | Triples in the test subset | | |
|---|---|---|---|---|---|---|---|
| 1 | *Memorization* | Steve Carell | **friend** | Stephen Colbert | Steve Carell | **friend** | Stephen Colbert |
| 2 | *One direction unseen* | Paul McCartney | **peer** | Michael Jackson | Michael Jackson | **peer** | Paul McCartney |
| 3 | *Both directions Unseen* (from test set) | Idaho | **adjoins** | Utah | Idaho / Utah | **adjoins** / **adjoins** | Utah / Idaho |
| 4 | *Asymmetry subset* | Warren Beatty | **gender** | Male | Male | **gender** | Warren Beatty |

Figure 5: Examples from FB15K-237 for each behavioral test to determine the symmetric relation capability.

For the ROTATE model, we use 256 negative per positive samples for the negative sampler, a learning rate of $5e^{-5}$ with the ADAM optimizer and a training and evaluation batch sizes of 1,024 and 16, respectively. We either fix the dimensions at 1,000 or the epochs at 500, similarly to the DISTMULT trials. For both KGE models, we run 5 independent experiments for every set of hyper-parameters, we then report the average performance.

For the experiments of COMPLEX, we use the official source code of the ROTATE model, where the best model hyper parameters are reported as follows: batch size 1024, negative sample size 256, hidden dimensions of 1000, $\gamma$ 200, $\alpha$ 1.0, learning rate 0.001, maximum steps of 100000 and a test batch size of 16.

For the HAKE model, we use the recommended configurations in the paper, which are also provided with the source code, with a batch size of 1024, negative sample size 256, hidden dimensions 1000, $\gamma$ 9.0, $\alpha$ 1.0, learning rate 0.00005, maximum steps 100000, test batch size 16, and their custom parameters of modulus and phase weights of 3.5 and 1.0 respectively.

For HYPERKG, the negative sample size is 5, $\lambda$ is 0.8, $\gamma$ is 0.5 and $\beta$ is n/2, with n=100.

For LINEARE, $\alpha$ is 0.5, $\beta$ is 1.0, $\gamma$ is set to 12, the embedding dimensions are 1000, a batch size of 2048 and a negative sample size of 128.

## Appendix D. Behavioral Tests

### D.1 Capability 1: Relation Symmetry

Examples in Figure 5 demonstrate the type of triples that are considered for the relation symmetry tests.

### D.1.1 TEST SET STATISTICS

The symmetry tests have the following number of triples:

| Test 1 | Test 2 | Test 3 | Test 4 |
|---|---|---|---|
| 5,70 | 1,308 | 226 | 3,000 |

The original Test Set contains 20,466 triples that represent a variety of relations, both symmetric and asymmetric.

### D.1.2 ADDITIONAL RESULTS

In the main paper, in Figure 2, we provide the results on the symmetry tests for varying the embedding dimensions for DISTMULT and ROTATE. In Figure 6a, we provide the results for varying the epochs.

## D.2 Capability 2: Entity Hierarchy

### D.2.1 TEST SET STATISTICS

The hierarchy tests have the following number of triples:

| Level 1 | Level 2 | Level 3 | Level 4 | Level 5 | Level 6 |
|---------|---------|---------|---------|---------|---------|
| 3,628   | 7,939   | 3,408   | 6264    | 21      | 1       |

Triples from Level 5 and Level 6 were filtered out because of the small data set size. The original Test Set contains 20,466 triples that represent a variety of relations with tails of different levels in the entity type hierarchy.

### D.2.2 ADDITIONAL RESULTS

In the main paper, in Figure 4, we provide the results on the hierarchy tests for varying the Epochs for DISTMULT. In Figure 6b, we provide the results for varying the embedding dimensions, and in Figure 7, we provide the same results for ROTATE

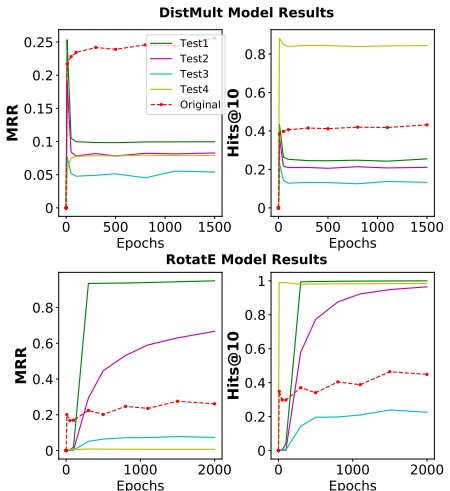

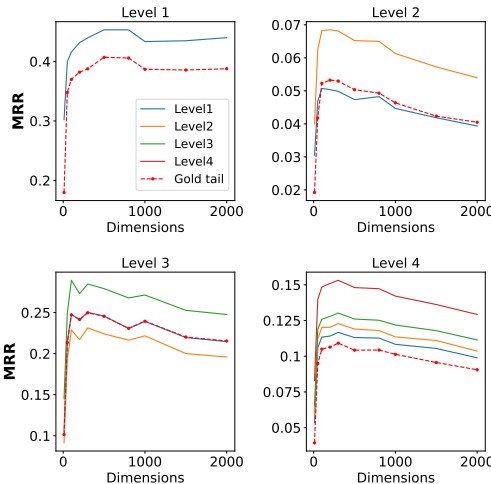

(a) MRR and Hits@10 results of RotatE and DistMult at different epochs, on the original test set and the hierarchy test sets.

(b) MRR results of DistMult at different Dimensions, on the original test set and the hierarchy test sets.

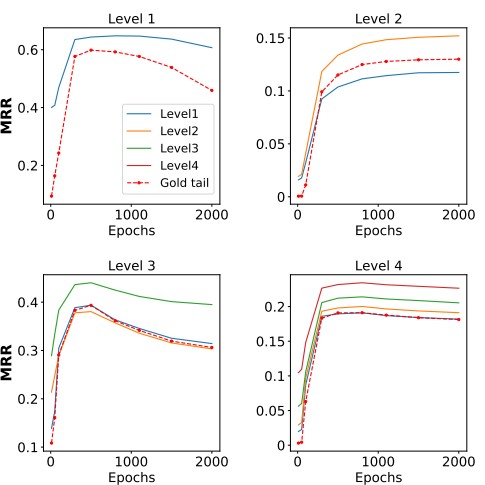

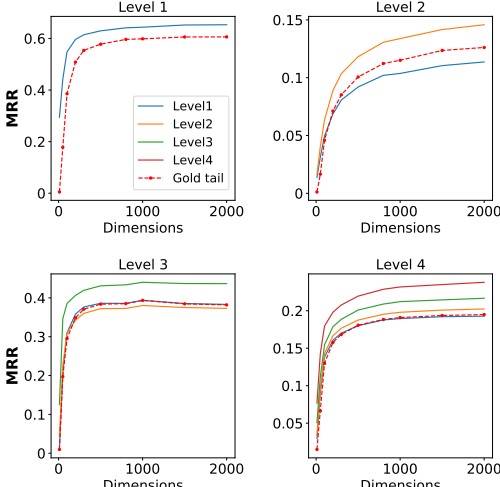

(a) MRR results of RotatE at different epochs, on the original test set and the hierarchy test sets.

(b) MRR results of RotatE at different epochs, on the original test set and the hierarchy test sets.

Figure 7: RotatE results for the hierarchy tests. Type constraints are most useful at the selected level of the test set

