# OpenReview forum: "Behavioral Testing of Knowledge Graph Embedding Models for Link Prediction"
_AKBC.ws/2021/Conference — AKBC 2021_

### Official Review · Reviewer_EYFt · 2021-07-18
**An interesting comparative study and benchmark for in-depth evaluation of knowledge-graph embeddings**

**Rating:** 8
**Confidence:** 4

**Review:**

This work has a twofold contribution: (1) Providing a new, more comprehensive evaluation benchmark for Knowledge Graph Embedding (KGE) models, (2) Comparatively evaluating numerous state-of-the-art KGE models with the proposed benchmark method

Overall this is work, alongside the benchmark to made publicly available, is well justified and seems quite important to the growth of this research field.

Strong Points:

S1 The need for a new KGE benchmark is indeed important and well justified. The traditional link prediction evaluation is not comprehensive enough

S2  The comparative evaluation reveals useful insights regarding existing KGE models, and even better — it shows what these KGE models actually capture, and what they do not. For example, all models fail in predicting super-categories (I.e.  Level 2 Types)

S3 The benchmark will be made publicly available and can be used by future KGE modles.

Weak Points:

W1 While the paper suggests multiple behavioral tests, only the first two are discussed and experimented with.

W2 The second test for including types in the training is somewhat problematic:
(A) First, this test seems somewhat unrelated to the goal of the paper, as it does not compare between the models, but checks a general hypothesis (whether it is useful to include type relations in the training data).
(B) Second, the results are not entirely convincing,  as it seems that the MRR are computed for a different test size —  it contains only entities from the same type, rather than all entities. I believe the space should have been reserved for an additional comparative evaluation between the model.

---

> ### Author Response · Authors · 2021-07-28
> **Thanks for your review, we plan to add more tests in the future**
>
> Thank you very much for your review!
>
> Regarding the two weak points:
> W1: We agree that adding more tests would be beneficial. We plan to add more tests in the future and hope that other researchers will also contribute. Nevertheless, we think that the initial two tests we picked serve as good examples to motivate the evaluation framework.
>
> W2: Thanks for pointing this out. We agree that a comparative evaluation between models would fit the narrative better. Originally we thought it might be interesting to highlight that the tests could also be used for other use cases. We have adjusted this by motivating Test 2 in the following way: "In the second test we explore how much model performance could be improved if the model had learnt to associate a relation with the correct entity type." We would be happy about feedback if this fits the narrative better.
>
> Since the submission, we have also defined an additional test: It tests how good models are at predicting the correct tail type for a given (head, relation, ?) query. For example, given the query (Stephen Hawking, BirthPlace, ?), we check if the model has learnt that the tail should be of type Place. Experiments for this test are still ongoing and we will add our findings in the camera-ready version upon acceptance.

---

### Official Review · Reviewer_EqfL · 2021-07-20
**An insightful proposal for adopting a different perspective on KGE evaluation.**

**Rating:** 7
**Confidence:** 4

**Review:**



This paper proposes a method for performing behavior testing for KG evaluation. Automated metrics are known to misrepresent the actual performance of the systems (see [1] for a critique of automated metrics for text generation). Additionally, automated metrics might only capture a subset of desired properties that a system is expected to have. Thus, any effort that aims to rethink evaluation strategies should be welcome by the community.


In that light, the work is timely and addresses an important problem. The authors take inspiration from behavioral testing (a specific type of unit test) and propose a series of capabilities (test type) for evaluating link prediction models trained on popular knowledge bases. The authors propose a rich list of such "capabilities" and experiment with two important subsets that test symmetry and entity hierarchy. Their experiments reveal meaningful insights about specific strengths and weaknesses of popular knowledge-graph embedding  (KGE) systems.


The most significant contribution of this work is conceptual; moving from summary to a more specialized set of metrics will help guide research and avoid situations where minute details (e.g., the tie-breaking strategy) has a significant effect on the results.

With that said, I am not entirely sure if labeling the proposed capabilities (Section 3.1, 3.2) as behavioral tests is quite correct, as they more closely resemble a benchmark (hence my slightly lower score). The difference is not merely semantic; the use cases are different: a test suite is essentially a checklist that can help ensure that a system performs as advertised (so it can be a set of binary criteria/assert statements). However, a benchmark supplies a way to compare and contrast two systems (typically associated with a metric that can be used to rank systems). A discussion in the final version would help position the paper and guide the future direction.


Strengths:

* The work addresses a significant problem and presents a reasonable attempt at solving it.
* The paper is well written and covers representative KGE systems.

Weaknesses:

* No weaknesses per se, but the authors should attempt to strengthen their proposal for capability 2 (entity hierarchy). Specifically, the details on why level 2 is worse than the other levels could lead to some interesting insights. Perhaps some spot-checking (looking at 100 random samples) could help.

* It might be helpful to give meaningful names to specific tests. For example, Test 1 could be called "overfitting check."



[1] Gehrmann, Sebastian, Tosin Adewumi, Karmanya Aggarwal, Pawan Sasanka Ammanamanchi, Aremu Anuoluwapo, Antoine Bosselut, Khyathi Raghavi Chandu et al. "The gem benchmark: Natural language generation, its evaluation and metrics." arXiv preprint arXiv:2102.01672 (2021).

---

> ### Author Response · Authors · 2021-07-28
> **Thanks for your review, we have added a discussion of behavorial tests vs benchmarks to the paper (see last paragraph on page 5)**
>
> Thank you very much for your review!
>
> Regarding the point of behavorial tests vs. benchmarks. Thanks for pointing out the distinction. We adopt the term behavorial test to follow Ribeiro et al. ACL 2020, who proposed the method for NLP systems and explored both directions: 1) comparing systems with each other and 2) defining a failure rate for a particular system. We have added a discussion of these two possibilities to the paper (see last paragraph on page 5).
>
> Regarding the weaknesses:
>
> Regarding level 2, thanks for the suggestion. We have looked into this further and found that the drop of performance in level 2 compared to the other levels likely stems from the type of relation. The relations that occur in level 2 rarely occur in other levels and most of the relations expect a tail of the level 2 type Person. We conjecture that the models are particularly bad at predicting in this scenario because of the large number of entities that have the type Person (4950; which constitutes 34% of all the entities in the set). We have also added this to the paper (see "Results" in Section 3.2)
>
> Since the submission, we have defined an additional test: It tests how good models are at predicting the correct tail type for a given (head, relation, ?) query. For example, given the query (Stephen Hawking, BirthPlace, ?), we check if the model has learnt that the tail should be of type Place. Experiments for this test are still ongoing and we will add our findings in the camera-ready version upon acceptance.
>
> Regarding naming the tests, when we define the tests (e.g. page 7), we have given them names (for Test 1 we have called it “Memorization”). We also repeat these names when a test number is first mentioned in the results section. Does this help or do you have any additional suggestions?

---

### Official Review · Reviewer_seHH · 2021-07-23
**Good direction for improving the evaluation methods on link prediction**

**Rating:** 8
**Confidence:** 4

**Review:**

This work proposes to apply behavioral testing in the evaluation of KGE models for link prediction. The authors first pointed out the drawbacks in the current evaluation method which typically averaging the accuracy metrics on all the triples in the test set. Then they suggested to apply the behavioral test evaluates the system's performance on different capabilities of the system. As two examples, the authors conducted such evaluation for symmetric relations and type hierarchies and further exposed discrepancy in the original evaluation.
Strength:

(1) This work pointed out the existing problems in the evaluation of KGE models, which makes the community focusing optimizing the averaged score on the benchmark without understanding the true properties of these systems. The proposed remedy and the two examples showed a good direction for progress.

(2) The comparison of different KGE models under different conditions in experiment 1 is insightful and showed that many of the models doesn't actualy generalize well to the truly unseen cases and the ranking on these unseen cases can change quite a lot comparing to the original evaluation. This further showed the danger of the current evaluation methods. The observation that a model is either good at recognizing symmetry or asymmetry also points out some potential issues in the current models.

Weakness:

(1) Although the paper proposes the improvement in the name of behavior testing, the main lesson seems to be enriching the evaluation to move from one averaged number to a set of meaningful tests each evaluating different aspects of the system or under different conditions. So it seems the focus should be on the variety of the evaluations instead of whether it is a black box behavioral test.

(2) It would be better if there are more capabilities tested beyond the two included.

---

> ### Author Response · Authors · 2021-07-28
> **Thanks for your review, we added a discussion on what the tests can be used for (see last paragraph on page 5) and plan to add more tests in the future**
>
> Thank you very much for your review!
>
> Regarding the two weaknesses:
> (1) Thanks for the feedback. We adopted the term behavorial testing to follow previous work Ribeiro et al. ACL 2020, which inspired ours. There the idea of evaluating with different tests under different conditions was termed as behavorial testing and we followed this terminology. We have also added a brief discussion on how the tests can be used (see last paragraph  on page 5).
>
> (2) We agree that adding more tests would be beneficial and we plan to add more tests in the future and hope that other researchers will also contribute. Nevertheless, we think that the initial two capabilities we picked serve as good examples to motivate the evaluation framework.
>
> Since the submission, we have also defined an additional test: It tests how good models are at predicting the correct tail type for a given (head, relation, ?) query. For example, given the query (Stephen Hawking, BirthPlace, ?), we check if the model has learnt that the tail should be of type Place. Experiments for this test are still ongoing and we will add our findings in the camera-ready version upon acceptance.

---

### Decision · Program_Chairs · 2021-08-18

**Decision:**

Accept

**Comment:**

This paper proposes a new and comprehensive evaluation benchmark for knowledge graph embedding methods and compares a set of existing knowledge graph embedding methods under this framework. All the reviewers agree that the proposed evaluation framework provides a more systematic way to evaluate the performance of knowledge graph embedding methods and is a good contribution for the community.